# Atrial Fibrillation and Ischemic Stroke despite Oral Anticoagulation

**DOI:** 10.3390/jcm12185784

**Published:** 2023-09-05

**Authors:** Roberto Galea, David Seiffge, Lorenz Räber

**Affiliations:** 1Department of Cardiology, Inselspital, Bern University Hospital, University of Bern, 3010 Bern, Switzerland; roberto.galea@insel.ch; 2Department of Neurology, Inselspital, Bern University Hospital, University of Bern, 3010 Bern, Switzerland; david.seiffge@insel.ch

**Keywords:** ischemic stroke, breakthrough strokes, OAC failure, stroke under OAC, left atrial appendage closure

## Abstract

Patients with atrial fibrillation (AF) experiencing ischemic stroke despite oral anticoagulation (OAC), i.e., breakthrough strokes, are not uncommon, and represent an important clinical subgroup in view of the consistently high risk of stroke recurrence and mortality. The understanding of the heterogenous potential mechanism underlying OAC failure is essential in order to implement specific therapeutic measures aimed at reducing the risk of recurrent ischemic stroke. However, due to the incomplete comprehension of this phenomenon and the limited available data, secondary stroke prevention in such high-risk patients represents a clinical dilemma. There are several available strategies to prevent ischemic stroke recurrence in AF patients with breakthrough stroke in the absence of competing causes unrelated to AF, and these include continuation or change in the type of OAC, addition of antiplatelet therapy, left atrial appendage closure, or any combination of the above options. However, due to the limited available data, the latest guidelines do not provide any specific recommendations about which of the above strategies may be preferred. This review describes the incidence, the clinical impact and the potential mechanisms underlying OAC failure in AF patients. Furthermore, the evidence supporting each of the above therapeutic options for secondary stroke prevention and the potential future directions will be discussed.

## 1. Introduction

Stroke is a leading cause of death and morbidity [1]. Atrial fibrillation (AF) is responsible for approximately 20–30% of ischemic strokes, especially in older patients [2,3]. Long-term oral anticoagulation (OAC) represents an effective treatment for preventing stroke in AF patients. Direct oral anticoagulants (DOAC), including both factor Xa inhibitors and direct thrombin inhibitors, are preferred over vitamin K antagonists (VKA) for primary and secondary stroke prevention in non-valvular AF [4]. However, patients with AF may still suffer from an ischemic stroke despite taking OAC. Randomized clinical trials (RCT) showed failure of OAC in a small but sizable (1–2% per year) percentage of non-valvular AF patients [5,6,7,8,9]. Similar rates were confirmed in real-life studies [10] and a meta-analysis [11]. Furthermore, according to recent observational data, up to one-third of strokes in AF patients occur despite OAC [12,13,14]. Emerging evidence from prospective observational cohorts [12,15,16] and post hoc analysis of the pivotal RCT comparing DOACs with VKA in patients with AF [17] found patients with breakthrough strokes despite OAC to be at increased risk of further recurrence constituting a highly vulnerable patient population. The optimal prevention strategy to further reduce recurrence risk in such patients is still unknown [15,18,19]. Breakthrough strokes may be caused by competing, non-AF-related mechanisms (i.e., large or small vessel disease), which accounts for approximately 30% of all breakthrough strokes or problems with anticoagulation [19]. In general, several therapeutical options have been reported: continuing with OAC, switching to another OAC, adding antiplatelet therapy, performing left atrial appendage closure (LAAC), or a combination of the above strategies [20,21].

This review describes the incidence, prognosis, and potential mechanisms underlying OAC failure and the available strategies to approach this clinical scenario. Finally, we also aim to evaluate the evidence related to the therapeutic options and the potential future directions.

## 2. Failure of Anticoagulation Therapy to Prevent Ischemic Stroke

The OAC medications currently recommended by European and American guidelines for preventing stroke in non-valvular AF patients include DOAC and VKA [4,22].

VKA represented the principal medical strategy for stroke prevention in patients with AF prior to the advent of DOAC. A meta-analysis summarizing the evidence of VKA compared with placebo and antiplatelet therapy in 28,044 patients with AF and a mean follow-up of at least 3 months showed that VKA with an achieved international normalized ratio (INR) between 2.0 to 2.9 reduced the risk of stroke by 64% as compared to placebo and by 39% as compared to single antiplatelet therapy [11]. However, a small percentage of patients under VKA (approximately 2.2%) experienced ischemic stroke [11]. Similar OAC failure rates have been reported in large real-life studies with VKA [23]. Evidence suggest that the majority of cerebrovascular events in AF patients under VKA might be related to subtherapeutic INR [24]. The narrow therapeutic index requiring frequent monitoring and dose adjustments in addition to the interaction with other drugs and diet contributed to the underuse of VKA and to the research of alternative pharmacological strategies.

Four large-scale RCTs assessed the efficacy and safety of dabigatran, rivaroxaban, apixaban, and edoxaban compared with VKA for prevention of thromboembolic events in patients with non-valvular AF [5,6,7,8]. An individual patient data meta-analysis including 71,683 patients enrolled in the 4 abovementioned RCTs reported that DOAC compared with VKA reduce the risk of thromboembolic events by approximately 20%, all-cause mortality by 10%, and intracranial hemorrhage by 50%, as compared with VKA [9]. Despite the significant reduction in both thromboembolic events (relative risk [RR]: 0.81, 95% confidence interval [CI]: 0.73–0.91; *p* < 0.001) and mortality (RR: 0.90, 95%CI: 0.85–0.95; *p* =< 0.001) as compared to VKA, a rate of 2.27 of ischemic strokes per 100 patients years was observed in patients randomized to DOAC (Table 1) [9]. Similar DOAC failure rates were observed in large cohort studies [23].

The latest European and American guidelines for management of AF recommend DOAC in preference to VKA for stroke prevention in selected AF patients (excluding patients with mechanical heart valves or moderate-to-severe mitral stenosis) with increased stroke risk [4,22]. However, no recommendations have been given on patients experiencing stroke despite OAC, except for the improvement of cardiovascular risk factors control. This guidance gap is particular important considering that this clinical subgroup represents up to one-third of all patients with AF hospitalized due to ischemic stroke and that these patients are at a particularly high risk of recurrent ischemic strokes [15,17,19]. A very recent individual participant data meta-analysis including 1163 AF patients enrolled in 5 pivotal RCTs (testing antithrombotic therapy for stroke prevention in AF) and experiencing a first post-randomization ischemic stroke while on study medication, showed a mortality rate of 12.4% at 3 months after stroke and an ischemic stroke recurrence of 6.2% at 1 year [17]. A retrospective multicenter observational study including 2946 consecutive AF patients hospitalized in a stroke center and discharged with diagnosis of stroke under OAC consistently showed very high rates of mortality (22.8%) and recurrent ischemic stroke (4.6%) at 3 months after the index event [19]. Another recent individual patient data meta-analysis including 5413 patients with AF and recent cerebral ischemia prospectively collected in observational studies reported that patients with as compared to those without OAC prior to index event had a 60% higher risk of ischemic stroke (adjusted hazard ratio [HR]: 1.6; 95%CI = 1.2–2.3; *p* = 0.005) after 6128 patient-years of follow-up [15].

Overall, AF patients with breakthrough stroke despite OAC are not uncommon in clinical practice and are associated with a particularly poor prognosis.

## 3. Identification of Mechanisms Underlying Stroke despite Oral Anticoagulation

The mechanism underlying ischemic stroke in patients with AF under OAC is still incompletely understood. The studies so far conducted with the aim of detecting potential predictors of ischemic stroke despite OAC were few and small [15,25,26], and showed controversial results, most likely due to the heterogeneous group of patients involved [21].

A large individual patient data pooled analysis of 7 prospective cohort studies including 5413 AF patients with recent cerebral ischemia showed that patients with as compared to those without OAC at the time of the ischemic event were older (median 79 vs. 77 years; *p* < 0.001), more frequently had hypertension (85.9% vs. 72.3%; *p* < 0.001), dyslipidemia (42.7% vs. 37.3%; *p* = 0.002), diabetes mellitus (37% vs. 21.7%; *p* < 0.001), or a history of stroke (38.4% vs. 19.2%; *p* < 0.001) [15]. The different cardiovascular risk in the presence of an apparently similar thromboembolic risk (the median CHA2DS2VASc score was 5 in both groups, although this score beyond a certain value might not adequately reflect thromboembolic risk) in the two study groups suggests that some strokes due to OAC failure may be attributed to non-cardioembolic mechanisms (atherosclerosis or small-vessel disease) [15]. Consistently, a multicenter American registry including AF patients experiencing an ischemic stroke showed that patients with vs. without INR ≥ 2 at the time of the index event were more likely to have ipsilateral moderate to severe arterial stenosis and a small infarct volume (<10 mL) [26]. Again, another multicentre observational study including 713 cerebrovascular ischemic events in patients with non-valvular AF showed that in more than one-third of cases, the ischemic strokes had a competing non-cardioembolic mechanism [25]. However, in the same study, off-label low-dose DOAC, atrial enlargement, high CHA2DS2-VASc score, and increased AF burden were all associated with ischemic stroke despite DOAC therapy, suggesting that both insufficient OAC and cardioembolic stroke might favor ischemic events occurrence [25].

Therefore, based on the above data, the potential mechanisms underlying ischemic strokes despite OAC might be classified into the following three categories [21]:(1)Ischemic stroke not related to AF, i.e., other stroke subtypes;(2)Insufficient OAC;(3)Cardioembolic stroke despite adequate OAC.

Each of the above mechanisms will be discussed below. Then, a diagnostic workflow suggesting how to approach patients with ischemic stroke under OAC will be described.

### 3.1. Ischemic Stroke Not Related to Atrial Fibrillation

Cardioembolic strokes represent approximately one-fourth of all ischemic stroke in clinical practice [27]. The concurrent causes of ischemic stroke mostly include cryptogenic strokes (representing more than one-third of all ischemic strokes) and large or small-vessel atherosclerosis (corresponding to about one-third of all ischemic strokes) [27]. The above rates differ in patients with AF under OAC. In a multicenter experience of almost three thousand AF patients hospitalized in a stroke center due to stroke despite OAC, Polymeris et al. showed that a competing stroke mechanism that was non-AF-related was detectable in only one-fourth (24.2%) of all ischemic strokes [19]. Similar percentages were reported by other cohort studies [25]. Within this population subgroup, the most common competing mechanism was large artery atherosclerosis (60.6%) patients, followed by small vessel disease (26.3%) and other etiologies (approximately 15%) including coagulopathies, peri-interventional stroke, endocarditis, other cardio-aortic pathologies, cervical artery dissection, and vasculitis [19]. Of note, no significant differences were observed in the distribution of competing mechanisms among patients with DOAC vs. VKA therapy at the time of the index stroke. At three months after the ischemic event, patients with competing mechanisms had higher adjusted odds for recurrent ischemic stroke as compared with patients with cardioembolism despite sufficient OAC [19].

### 3.2. Insufficient OAC

In line with other previous studies [25], Polymeris et al. recently showed in a multicenter observational that insufficient OAC was observed in almost one-third (31.7%) of AF patients with stroke despite OAC [19]. In this study, insufficient OAC was defined as at least one of the following criteria: self-reported non-adherence (i.e., history of missing intake of anticoagulants within the last 3 days before the stroke); low OAC activity on admission (i.e., INR < 2.0 in VKA-treated patients; plasma level < 30 ng/mL in DOAC-treated patients); inappropriately low DOAC dose or dosing frequency (according to current product labelling). Some further scenarios that might be included in this subgroup are drug–drug interactions and food interactions (particularly important for rivaroxaban). Of note, some of the above criteria (e.g., an off-label low dose of DOACs) were independently associated with ischemic stroke recurrence in other multicenter observation studies (odds ratio [OR], 3.18; 95%CI, 1.95–5.85; *p* < 0.001) [25].

At three months after stroke, Polymeris et al. observed a similar rate of stroke recurrence in patients with stroke due to insufficient anticoagulation as compared to those with cardioembolism despite sufficient OAC [19].

### 3.3. Cardioembolic Stroke despite Adequate OAC

Cardioembolic stroke in AF patients submitted to adequate OAC represents a challenging scenario in clinical practice for several reasons. Firstly, a commonly accepted definition of this adverse event is not yet available. Some research groups proposed to consider as cardioembolic all those ischemic strokes occurring in AF patients under adequate OAC and in the absence of competing stroke mechanisms that are non-AF-related [19]. Secondly, the underlying mechanism is unknown. Paciaroni et al., in their multicenter observation study including AF patients under DOAC, observed an independent association between atrial enlargement, high CHA2DS2-VASc score, and increased AF burden with ischemic stroke [25]. This observation suggests that some recurrent ischemic strokes in AF patients under OAC may be related to more advanced atrial disease (e.g., increased AF burden and severe atrial enlargement) [21]. Thirdly, while in the previous two mechanisms the adoption of new therapeutical strategies might intuitively reduce stroke recurrence (i.e., improvement of cardiovascular risk factors control, medication adherence, etc.), in cardioembolic stroke under adequate OAC, a therapy change (except for switching to other OACs) does not intuitively seem an important factor in preventing the ischemic event.

Reassuringly, the prognosis of this stroke subtype seems similar to that related to insufficient OAC and better than that of concurrent mechanisms that are non-AF-related [19].

### 3.4. Diagnostic Workflow

The determination of the mechanism or of the combination of mechanisms underlying the breakthrough strokes is necessary to tailor the secondary stroke prevention strategy [12,15].

The recommended diagnostic work-up includes three main steps [21,28].

Firstly, a competing non-AF mechanisms of stroke should be identified. This diagnostic process includes brain magnetic resonance imaging (MRI) that allows evaluation of possible stroke mechanism based on the patterns (embolic, deep, large, or small scattered), distribution on diffusion-weighted imaging and presence of prior strokes on fluid-attenuated inversion recovery imaging [29]. Furthermore, vascular imaging of the brain-supplying arteries with computed tomography (CT)/MRI angiography or doppler ultrasound might be used to exclude ipsilateral high-grade stenosis. Following exclusion of lacunar stroke and atherosclerosis of large vessels with the above imaging exams, further diagnostic tools might be considered on an individual basis to investigate the remaining and uncommon potential stroke mechanisms as vessel wall imaging or fat-saturated imaging sequences (to detect dissection, unstable plaque, or vasculitis [29]), cerebrospinal fluid analysis (if vasculitis is suspected), imaging with CT angiography, transesophageal echocardiogram (TEE), and transcranial duplex sonography (if aortic arch disease or paradoxical sources of embolism are suspected), coagulation testing for hypercoagulability (if cancer or antiphospholipid antibody syndrome is suspected) [30].

The second step of the recommended diagnostic work should exclude a medication error. Information from patient and/or caregiver interviews should be carefully collected in order to verify medication adherence, the appropriateness of the DOAC doses, and to exclude concomitant drugs/food influencing metabolization and clearance of OAC (e.g., rifampicin, carbamazepine, phenytoin, etc.) [21]. While INR measurement proved to be an accurate test to monitor anticoagulant activity of VKA and a INR > 2 showed to provide an efficient protection against ischemic stroke, no analogs of similar accuracy have been so far reported for DOAC [11,21]. Calibrated anti-factor Xa activity, a DOAC-specific coagulation assay, is difficult to interpret as there are no target levels and the absolute values expected at different time points after last DOAC intake are very heterogeneous [31]. As recently suggested, anti-factor Xa activity should be used to determine non-adherence only in the event of very low levels (<30 ng/mL) being observed on admission, i.e., a value suggesting no direct factor Xa inhibitor was used in the previous 48 h [21]. Although routine assessment of plasma levels is currently not a standard of care for all patients with breakthrough strokes despite DOAC therapy, these measurements may inform etiological work beyond reported compliance by patients and/or next of kin.

Finally, the diagnostic workflow should be completed by performing imaging exams (i.e., TEE) to exclude intracardiac sources of embolic stroke, such as intracardiac thrombus, endocarditis, tumors, and evidence of severe atrial dysfunction (e.g., LA enlargement) or of thrombogenic LAA [32].

## 4. Secondary Prevention of Ischemic Stroke despite Oral Anticoagulation

Secondary stroke prevention in AF patients experiencing ischemic cerebrovascular event despite OAC is challenging. There are several reasons for this, including the incomplete understanding of the mechanisms underlying the ischemic events and the heterogeneous and high-risk population treated. As reported above, the initial care should firstly focus on excluding alternative reasons for stroke and those cases where insufficient OAC (including both reduced patient compliance and OAC underdosing) was the most likely reason of the ischemic event. However, this patient subgroup, although it might be easily treated by optimization of their OAC therapy, represents only a minority (31.7%) of patients with a breakthrough stroke [19]. It is also unclear whether patient education and other interventions to improve compliance may reduce the risk of recurrence in this patient population. The latest American Heart Association/American Stroke Association guidelines recommend tailoring the secondary stroke prevention strategy according to the suspected stroke etiology and regardless the antithrombotic therapy ongoing at the moment of stroke [27]. However, no specific recommendations have been given by the American Cardiology and Neurology Society guidelines about how to prevent further ischemic events in the specific scenario with breakthrough stroke [22,27]. Several proposed strategies have been reported so far. Polymeris et al. showed that in a multicenter observational study of almost three thousands AF patients hospitalized in a stroke center due breakthrough stroke, a majority of cases (almost 85%) secondary stroke prevention entailed OAC continuation (including switch to another OAC type) whereas addition of antiplatelet therapy (in 12.8%) or left atrial appendage closure ([LAAC] in 1%) were considered only infrequently [19]. Available supporting evidence and limitations of each available strategies will be discussed below (Figure 1).

### 4.1. OAC Continuation

OAC is an efficient therapy for secondary prevention of ischemic stroke in patients with AF [33]. However, the clinical impact of OAC re-initiation timing on the risk of stroke recurrence and hemorrhagic transformation after an acute ischemic stroke was only recently studied, as in the phase III DOAC trials recent stroke was an exclusion criteria and since the few RCTs conducted on this topic had relevant limitations [38]. In the ELAN (early vs. late initiation of direct oral anticoagulants in post-ischemic stroke patients with atrial fibrillation) study [34], 2013 AF patients with recent ischemic stroke were randomly allocated to receiving in a 1:1 fashion either an early (within 48 h after a minor or moderate stroke or on day 6–7 after a major stroke) or later OAC (day 3–4 after a minor stroke, day 6–7 after a moderate stroke, or day 12–14 after a major stroke). The primary endpoint was the composite of recurrent ischemic stroke, systemic embolism, major extracranial bleeding, symptomatic intracranial hemorrhage, or vascular death within 30 day after the index event. Patients with hemorrhagic infarction were excluded from this trial. The primary endpoint occurred in 2.9% of patients receiving early and 4.1% of patients receiving later OAC (risk difference, −1.18 percentage points, 95%CI, −2.84 to 0.47). At 30 days, recurrent ischemic strokes were reported in 1.4% and 2.5% (odds ratio: 0.57; 95%CI, 0.29 to 1.07) of early and late groups, respectively. No difference in intracranial hemorrhage was noted (in both groups only two events were observed). Although in this trial no statistical hypothesis was tested, the provided risk estimates encourages early initiation of OAC [34].

In breakthrough strokes with no evidence of underdosing or malcompliance, a change in OAC type is often practiced. In particular, dabigatran 150mg twice daily is frequently preferred over others since in trials directly comparing DOACs with VKA it was the only drug reportedly superior in terms of stroke prevention as compared to VKA [5]. Notwithstanding these results, there is no available evidence to support this practice [12,15,18], i.e., no comparative trials confirmed superior efficacy of dabigatran 150mg in head-to-head trials against other DOACs. In addition, the Rely trial was criticized for the inclusion of a very low-risk AF population [5]. In a large retrospective national cohort including 2908 patients admitted to public hospitals in Hong Kong due to ischemic stroke despite DOAC, Bonaventure et al. observed that switching to another OAC was associated with an increased risk of stroke recurrence at 6 years after the index stroke (in the VKA switch group: adjusted HR; 1.96; 95%CI: 1.27–3.02; *p* = 0.002; in the DOAC switch group: adjusted HR: 1.62; 95%CI: 1.25–2.11; *p*-value < 0.001), as compared to the subgroups of patients who continued the same therapy [18]. Consistently, other multicenter observational studies did not show any benefit in terms of recurrent stroke reduction with OAC switching [19]. Overall, continuation of OAC therapy is still the standard secondary prevention strategy with a better level of evidence as compared to alternative strategies. Based on the available studies, the routine switch to other OAC medication does not seem justified unless particular clinical scenarios are present, such as reduced adherence related to a specific drug aspect (costs, number of pills, side effect), drug interactions, labile INR, concurrent antiphospholipid antibody syndrome, mechanical valves, hemodynamically significant mitral valve stenosis, or left ventricular thrombus.

### 4.2. Addition of Antiplatelet Therapy

Evidence suggests that some breakthrough strokes may be related to non-cardioembolic mechanisms, where OAC may not be an important factor in preventing the ischemic event. The addition of antiplatelet therapy is another therapeutic option, which has so far not been studied in a dedicated RCT including breakthrough strokes. Dentali et al. tested the complementary effect of antiplatelet and anticoagulant therapy in AF patients in a meta-analysis including ten RCTs comparing aspirin plus OAC with OAC therapy alone in patients with at least 3 months of follow-up. The authors observed similar thromboembolic event rates in more than 4000 patients for AF patients receiving combined aspirin-OAC therapy compared with OAC therapy alone. As expected, the rate of major bleeding was higher in patients receiving combined therapy compared with OAC therapy alone [39]. Similar results were shown by another meta-analyses derived from a general population with AF under OAC. The authors observed that use of antiplatelet therapy paradoxically increased the risk of stroke modestly (relative risk [RR]: 1.33; 95%CI: 0.98–1.79) and, as expected, also increased the risk of major bleeding (RR: 1.54; 95%CI: 1.35–1.77) including intracranial hemorrhage (RR: 1.64; 95%CI: 1.20–2.24) [40]. Consistently, studies including AF patients with breakthrough strokes showed that the addition of antiplatelets to anticoagulants was associated with higher odds for the stroke recurrence [18,19].

Collectively, the routine addition of antiplatelet therapy following breakthrough stroke may not be recommended, except possibly for short-term use in patients with arterial embolism from unstable atherosclerotic plaques [21]. However, dedicated RCTs are required to better understand if and in which AF patients the addition of antiplatelet therapy on top of OAC might reduce ischemic stroke recurrence.

### 4.3. Left Atrial Appendage Closure

Postmortem and echocardiographic studies have shown that the vast majority of all cardiac thrombi in patients with AF originate from the left atrial appendage (LAA) [41]. Thus, LAAC, consisting of exclusion of the LAA cavity from the circulation by implanting a cardiovascular device at LAA ostium, has been established in clinical practice as an attractive alternative to OAC for preventing stroke in AF patients [4,22]. A meta-analysis including 1114 AF patients randomized to either LAAC or VKA showed a similar incidence of composite of stroke, cardiovascular death, or systemic embolism (HR: 0.82; 95%CI: 0.58 to 1.17; *p* = 0.27) and lower rates of mortality (HR: 0.73; CI: 0.54–0.98; *p* = 0.035) and haemorrhagic stroke (HR: 0.20; CI: 0.07–0.56; *p* = 0.002) in LAAC as compared to VKA groups [42] at a follow-up of 4343 patient-years. However, the advent of DOACs and their widespread use in clinical practice as first-line therapy for stroke prevention in AF led international guidelines to recommend LAAC only in selected AF patients with contraindication to long-term OAC [4,22,43,44]. However, no specific recommendations were given about performance of LAAC for preventing stroke recurrence in AF patients experiencing cerebrovascular ischemic events despite OAC. The available evidence related to the benefit of LAAC in such high-risk patients is based on a few small observational studies [35,36,45,46] (Table 2). The largest one is a sub-study of the Amplatzer Cardiac Plug multicenter registry including 115 AF patients undergoing LAAC following a breakthrough stroke [35]. After a mean follow-up of 1349 patient years, the observed annual cerebrovascular ischemic events rate was 2.6% (a 65% relative reduction according to the CHA2DS2-VASc score) whereas the observed annual major bleeding rate was 0% (a 100% relative reduction according to the HAS-BLED score) [35]. Interestingly, the majority of patients we re discharged under either single (SAPT) or dual antiplatelet therapy (DAPT) whereas only a few patients (7.8%) were discharged under OAC [35]. Similarly, a recent prospective single-center observational study including 39 patients undergoing Watchman or Amulet implantation due to OAC failure and discharged under DAPT showed a reduction from scores predicted to observed annual rates of both thromboembolic (−14%) and bleeding (−100%) events [47]. On the other hand, Freixa et al. showed, in a retrospective analysis of 22 breakthrough stroke patients submitted to LAAC and continued OAC, a significant reduction in cerebrovascular events 2 years after the procedure as compared to the pre-LAAC events rates (0.1 ± 0.3 vs. 2.0 ± 1.0 events; *p* < 0.01) [36]. Reassuringly, the so far conducted observational studies on breakthrough stroke patients undergoing LAAC showed similar rates of technical success, procedure-related complications, and device-related thrombus as compared to other studies conducted on patients submitted to LAAC with other clinical indications [35,36,45,46]. A recently presented unpublished propensity score matching analysis comparing 433 patients receiving LAAC after thromboembolic event despite OAC with 433 patients continuing/switching OAC after breakthrough stroke showed a significantly reduced risk of the primary composite outcome in LAAC as compared to OAC groups (composite of recurrent stroke, systemic embolism, and cardiovascular death: HR 0.33; 95%CI 0.19–0.58) [48].

LAA thrombus can be observed in patients experiencing ischemic stroke despite OAC. This finding has been considered an absolute contraindication to LAAC for years, due to the potentially increased thromboembolic risk associated with the intervention and due to the exclusion of such patients from the pivotal RCTs comparing LAAC to VKA [42]. However, the procedural iteration occurred in the last few years in terms of procedural planning [49], procedural guidance [50], and devices used [51,52] led researchers to consider LAAC even in patients at higher risk, including those with LAA thrombus. We recently reported on a retrospective multicenter analysis of 53 clinically indicated LAAC procedures performed in patients with LAA thrombus encouraging procedural outcomes, including a technical success in 100% of patients, 0% of cardiac tamponade and only one primary endpoint (composite of stroke, systemic embolism, or cardiovascular death) event [53]. Of note, only a small percentage of patients (approximately 7%) had a history of recurrent thromboembolic events despite OAC, and an equally small percentage of patients (10%) were discharged under OAC [53].

The post-LAAC antithrombotic drug regimen to be prescribed in patients with breakthrough stroke is unclear. The studies so far conducted suggest that OAC [36,45], antiplatelet therapy [35], or both [46] might be considered at discharge after a successful LAAC procedure (Table 2). However, the relevant studies’ limitations, including the small size, the retrospective design, the lack of OAC adherence assessment, and the inherent selection bias do not allow us to draw any definitive conclusions. The limited evidence available on patients with breakthrough strokes undergoing LAAC suggest that either OAC alone or in addition to short-term aspirin or clopidogrel or DAPT for a few months followed by long-term SAPT may be appropriate strategies to prevent thromboembolic events after successful LAAC [43,44]. Sondergaard et al. reported a propensity score matching analysis including 1527 AF patients undergoing LAAC and discharged under OAC plus aspirin vs. antiplatelet therapy (91% on DAPT) and showed no differences between groups in terms of freedom from non-procedural thromboembolic conditions (98.8% vs. 99.4%; *p* = 0.089) at 6 months after procedure, although a higher DRT rate was observed in the antiplatelet group (3.1% vs. 1.4%; *p* = 0.014) [23]. Recently, LAAOS III reported the outcomes of a multicenter RCT testing the effect of additional surgical LAA exclusion in almost 5000 AF patients scheduled to undergo cardiac surgery for another indication. At a mean follow-up of 3.8 years, surgical LAAC on top of standard OAC reduced the risk of stroke or systemic embolism by 33% (4.8% vs. 7.0%; HR: 0.67; 95%CI: 0.53–0.85; *p* = 0.001) compared with no LAAC (OAC only) suggesting a synergistic benefit of surgical LAAC and OAC [54]. Based on these observations and on the high stroke risk of these patients, long-term OAC appears to be an appropriate therapy after LAAC for patients with breakthrough strokes. However, patients with breakthrough strokes seem to often have not only a very high ischemic risk but also a high bleeding risk, as recently reported by RENO-EXTEND study investigators (5.8% rate of major bleedings at a mean follow-up of 15.0 ± 10.9 months) [55], suggesting that a treatment regimen including DAPT or OAC/SAPT following LAAC may increase bleeding.

Collectively, the little available evidence suggests that LAAC might be an efficient strategy to further reduce the risk of stroke recurrence in patients with breakthrough stroke. However, further evidence is needed to better assess both the clinical benefit of LAAC and the ideal antithrombotic regimen following LAAC in such high-risk patients.

### 4.4. Rhythm Control

Rhythm control therapy is an emerging approach to potentially reduce stroke recurrence beyond the effect of anticoagulation alone in patients with AF. The evidence about the benefit of this strategy on reducing stroke risk in the overall AF population is still controversial [56]. However, very few studies have so far investigated the effects of early rhythm control in AF patients with recent ischemic stroke. The RAFAS (Risks and Benefits of Early Rhythm Control in Patients With Acute Strokes and Atrial Fibrillation: A Multicenter, Prospective, Randomized Study) trial compared a strategy of early rhythm control within 2 months after an acute stroke with standard care in a total of 300 patients [37]. Rates of stroke recurrence were lower at 12 months in the early rhythm control group (1.7%) compared with the standard care group (6.3%, *p* = 0.034). While antiarrhythmic drugs were used early following acute stroke (<10 days), pulmonary vein ablation was performed later during the study course (>3 months), and no safety signals were observed [37]. However, it is important to underline several aspects: first, patients older than 80 years, or with life expectancy of <12 months, or with large left atrium (diameter > 55 mm), or with contraindication to OAC, or with recent/planned acute coronary syndrome/cardiac intervention were excluded from the study; second, the study was not double-blinded and we cannot exclude that the early rhythm group received more aggressive drug treatment to prevent stroke recurrence; third, data regarding to the use of OAC at the time of the ischemic event were not reported [37].

As a conclusion, rhythm control following breakthrough strokes is a strategy that might be considered in patients without particularly high-risk characteristics. Clearly, further studies testing the impact of this strategy on stroke recurrence in patients with breakthrough strokes are required.

## 5. Future Perspectives

Regarding pharmaceutical approaches, a novel generation of anticoagulants, the direct factor XI/Xia inhibitors are currently being evaluated for stroke prevention in large phase 2 and phase 3 RCTs. Whether these compounds will be more efficacious as compared to current DOACs remains to be investigated. Due to their subsidiary role in hemostasis, it is assumed that factor XI/Xia inhibition can inhibit thrombus formation with lower hazard for bleeding. The two small molecule oral direct inhibitors of factor XIa, asundexian (OCEANIC-AF, NCT05643573) and milvexian (LIBREXIA-AF, NCT05757896) will be studied in two large phase 3 stroke prevention trials in the close future.

In light of the limited evidence supporting current strategies used for secondary prevention following breakthrough strokes, adequately powered RCTs are required to test the benefit of these therapeutic options. In the “Early closure of left atrial appendage for patients with AF and ischemic stroke despite anticoagulation therapy—the ELAPSE RCT”, a strategy of LAAC in combination with continued DOAC therapy will be compared to DOAC alone in at least 482 patients with breakthrough stroke (with an event-driven adaptive design). The ELAPSE study is a superiority trial with a primary composite endpoint including ischemic strokes, systemic embolism, and cardiovascular death assessed at a maximum of 4 years. In the ongoing Occlusion-AF trial (NCT03642509), patients with a recent cerebrovascular ischemic event (<6 months, independent of the therapy at the timepoint of stroke) will be receiving either LAAC or DOAC therapy, and the primary endpoint is a composite of stroke, systemic embolism, major bleeding, and all-cause mortality [57]. Finally, 4000 AF patients with high stroke risk (defined as a CHA2DS2VASc score of ≥4 with or without history of stroke) will be included in the LAAOS IV trial, a RCT comparing LAAC (with Watchman FLX implantation) on top of OAC with OAC alone in terms of thromboembolic events at a maximum of 3 years. The study is aimed at confirming the synergistic effect of OAC and LAAC in preventing ischemic stroke recently shown by LAAO III trial (but with a percutaneous instead of a surgical LAA closure) [54].

Other non-LAAC-based treatment strategies entail the use of a carotid filter. The INTERCEPT trial (NCT05723926) is assessing whether the use of bilateral carotid filter implants in addition to OAC will reduce the risk of stroke recurrence in AF patients with recent (<12 months) ischemic stroke. Early rhythm control is currently being tested in the STABLED trial (NCT03777631) that will compare in 250 AF patients with recent stroke catheter ablation on top of OAC vs. OAC alone in terms of a composite of recurrence of cerebral infarction, systemic embolism, all-cause death, and hospitalization for heart failure.

## 6. Conclusions

Patients with AF experiencing cerebrovascular events despite OAC are not uncommon in clinical practice and are associated with poor prognosis. The mechanism underlying OAC failure is still not completely understood making, the prevention of recurrent events a challenge. There are several available pharmaceutical and interventional strategies for prevention of stroke recurrence, and some of them promising; however, there is still little supporting evidence.

## Figures and Tables

**Figure 1 jcm-12-05784-f001:**
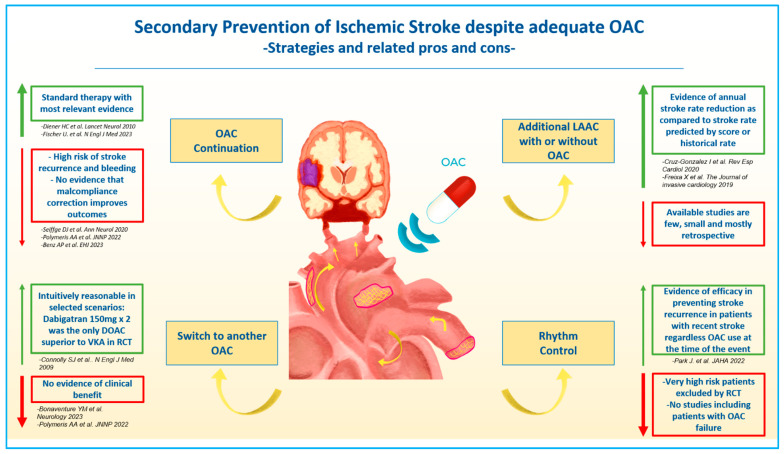
Evidence related to pharmacological, interventional, and combined strategies aimed at preventing ischemic stroke recurrence in patients with atrial fibrillation experiencing cerebrovascular ischemic events despite adequate oral anticoagulation. OAC, oral anticoagulant; LAAC, left atrial appendage closure; VKA, vitamin K antagonists; RCT, randomized clinical trial [5,15,17,18,19,33,34,35,36,37].

**Table 1 jcm-12-05784-t001:** Ischemic stroke rates reported in randomized clinical trials comparing direct oral anticoagulants and vitamin K antagonists for stroke prevention in patients with non-valvular atrial fibrillation.

Study Design	First Author/ Year of Publication	Patients No.	Age Median	CHADS2 Score Mean	History of Stroke/TIA/SE %	Mean Follow-Up Duration (y)	Treatment Arms		Ischemic Stroke Rate in Patients under DOAC %/Patient/Year	Ischemic Stroke Rate in Patients under VKA %/Patient/Year
RCT	Connolly SJ et al./2009 [5]	18,113	71.5 *	2.2	20 ¶	2	Dabigatran (150/110 mg)	VKA (INR 2–3)	1.34 in dabigatran 110 mg	1.20
									0.92 in dabigatran 150 mg	
RCT	Patel MR et al./2011 [8]	14,264	73	3.5	54.8	1.9	Rivaroxaban (20/15 mg)	VKA (INR 2–3)	1.7 µ	2.2 µ
RCT	Granger CB et al./2011 [7]	18,201	70	2.1	19.4	1.9	Apixaban (5/2.5 mg)	VKA (INR 2–3)	0.97	1.05
RCT	Giugliano RP et al./2013 [6]	21,105	72	2.8	28.3 ¶	2.8	Edoxaban (60/30 mg)	VKA (INR 2–3)	1.25 in edoxaban 60 mg	1.25
									1.77 in edoxaban 30 mg	

* Mean only stroke/TIA were included; ¶ Only stroke and TIA were included; µ stroke and systemic embolism were included; TIA, transient ischemic attack; SE, systemic embolism; DOAC, direct oral anticoagulant; VKA, vitamin K antagonists; RCT, randomized clinical trial.

**Table 2 jcm-12-05784-t002:** Available studies on percutaneous left atrial appendage closure performed in patients with atrial fibrillation experiencing an ischemic stroke despite oral anticoagulation.

First Author/ Year of Publication	Study Design	Patients No.	Age Mean	CHADS2 Score Mean	LAAC Device Implanted	Main Post-LAAC Antithrombotic Drug Regimen	Mean Follow-Up Duration (y)	Ischemic Stroke Recurrence Rate %/Patient/Year
Freixa X et al./2019 [36]	SC—RS	22	NA	NA	NA	OAC	1.8 *	2.5
Masjuan J et al./2019 [46]	MC—PS	19	72.1	5.3	ACP or Amulet	OAC + Aspirin	1.45	0
Cruz-Gonzalez I et al./2020 [35]	MC—RS	115	73.8	5.5	ACP	DAPT	1.35	2.6
Galloo X et al./2022 [45]	MC—RS	15	78.1	6	Amulet or Wachman	OAC	NA	2 events
Pracon R et al./2022 [47]	SC—PS	39	73 *	5 *	Amulet or Wachman	DAPT	1	7.7

* Median was reported instead of mean; LAAC, left atrial appendage closure; SC, single-center; MC, multicenter; RS, retrospective study; PS, prospective study; NA, not available; ACP, Amplatzer cardiac plug; OAC, oral anticoagulants; DAPT, dual antiplatelet therapy.

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
