# Peer review of "Atrial Fibrillation and Ischemic Stroke despite Oral Anticoagulation"

_jcm, 2023, doi:10.3390/jcm12185784_

Round 1

Reviewer 1 Report

I've read and appreciated the article entitled "Atrial Fibrillation and Ischemic Stroke despite Oral Anticoagulation".

It is well written, clear and exhaustive, even if something is lacking and should be added.

I would suggest to add/change:

line 30 add "expecially in older patients" because percentages are different at different ages

line 45: I would add "or problems with anticoagulation" as the authors themselves state later

Cap 3, line 140: add a subheading "incorrect use od DOAC in antiphospholipid antibody syndrome"

Add a subheading after 3.3: "antiphospholipid antibody syndrome" and explain that DOAC are not recommended expecially if double or triple positivity is diagnosed and cite the TRAPS study in references

I do not agree about rhytm control as a strategy to reduce embolism; the authors should cite thousands papers showing that it is not an option and not only the couple of papers hypotesing that it is

For the same reason I would delete rhytm control from the figure and replace it with "adding antiplatelet agent to anticoagulation" (always in the figure) with pros and cons

429 rhythm control paragraph: I would shorten it for the reasons explained; as alternative it should be enlarged explaining all the results against it as option for reducing embolism

RAFAS study should be explained better with all its limits. Not anly saying "However, it is important to underline the risk category (higher as compared to that of populations included in the above studies) of the population included in this 440 study: patients older than 80 years, or with life expectancy of <12 months, or with large 441 left atrium (diameter >55 mm), or with contraindication to OAC or with recent/planned 442 acute coronary syndrome/cardiac intervention were excluded from the study. Furthermore, data regarding to the use of OAC at the time of the ischemic event were not reported"

Author Response

1) I've read and appreciated the article entitled "Atrial Fibrillation and Ischemic Stroke despite Oral Anticoagulation". It is well written, clear and exhaustive, even if something is lacking and should be added. I would suggest to add/change:line 30 add "expecially in older patients" because percentages are different at different ages

We would like to thank the reviewer for his/her overall positive reception of our manuscript. We have added the suggested sentence.

2)…line 45: I would add "or problems with anticoagulation" as the authors themselves state later

We would like to thank the reviewer for this comment. We have corrected the text as suggested.

3)Cap 3, line 140: add a subheading "incorrect use of DOAC in antiphospholipid antibody syndrome". Add a subheading after 3.3: "antiphospholipid antibody syndrome" and explain that DOAC are not recommended expecially if double or triple positivity is diagnosed and cite the TRAPS study in references.

We would like to thank the Reviewer for this comment. We acknowledge that patients with antiphospholipid antibody syndrome at high risk represent a special condition of extreme interest that might be more extensively discussed in our review. However, since we decided to focus on the general topic of “patients with stroke under OAC”, we deliberately avoided describing in detail specific pathologies related to this clinical scenario in order to keep the focus of our readers on the main topic. In this regard, we wrote the following sentence:

Line 217: “coagulation testing for hypercoagulability (if cancer or antiphospholipid antibody syndrome is suspected”;

Line 309: “Based on the available studies, the routine switch to other OAC medication does not seem justified unless particular clinical scenarios are present: reduced adherence related to a specific drug aspect (costs, number of pills, side effect), drug interactions, labile INR, con-current antiphospholipid antibody syndrome, mechanical valves, hemodynamically significant mitral valve stenosis or left ventricular thrombus”.

We think that the above sentences are sufficient to mention the need of VKA in selected patients with stroke under OAC, including those with antiphospholipid antibody syndrome. However, if the reviewer strongly thinks this point needs to be discussed in a different way in this review, we remain entirely open for suggestions regarding how the text should be further modified.

4)I do not agree about rhythm control as a strategy to reduce embolism; the authors should cite thousands papers showing that it is not an option and not only the couple of papers hypotesing that it is. For the same reason I would delete rhythm control from the figure and replace it with "adding antiplatelet agent to anticoagulation" (always in the figure) with pros and cons….Line 429 rhythm control paragraph: I would shorten it for the reasons explained; as alternative it should be enlarged explaining all the results against it as option for reducing embolism. RAFAS study should be explained better with all its limits…

We would like to thank Reviewer for this comment. We acknowledge that benefit of rhythm control in reducing thromboembolic events in AF patients is still controversial. However, the evidence about the benefit of this strategy are indeed accruing (Camm AJ et al. J Am Coll Cardiol 2022;79:1932–1948). As reported by the Reviewer, mostly in the past, several studies failed to prove a relevant clinical benefit of rhythm control as compared to heart rate control strategy in AF patients: Hohnloser SH et al. Lancet. 2000;356:1789–1794; Wyse DG et al. N Engl J Med.2002;347:1825–1833; Van Gelder IC et al. N Engl J Med. 2002;347:1834–1840; Roy D et al. N Engl J Med. 2008;358:2667–2677; Ogawa S et al. Circ J. 2009;73:242–248. Carlsson J et al. Card Electrophysiol Rev. 2003;7:122–126). However, as suggested by Camm et al, it is possible that results from these trials (the vast majority of which was conducted more than 15 years ago) were affected by the lack of safe and effective therapies, small sample sizes, inclusion of patients with long-standing AF, variable endpoints, less stringent monitoring, and the use of inappropriately high or suboptimal antiarrhythmic drug doses, potentially reducing efficacy of rhythm control strategy.

On the other hand, several recent studies (Srivatsa UN et al.Circ: Arrhythm Electrophysiol. 2018;11:e005739 ; Saliba W et al. Heart Rhythm. 2017;14:635–642; Packer DL et al. JAMA. 2019;321:1261–1274; Friberg L et al. Eur Heart J. 2016;37:2478–2487; Kirchhof P et al. N Engl J Med. 2020;383:1305–1316; Hohnloser SH et al. N Engl J Med 2009;360:668-78) showed that active rhythm control of AF by methods such as catheter ablation, with appropriate anticoagulation, significantly reduce the risk of ischemic stroke. EAST-AFNET 4 Trial (Kirchhof P et al. N Engl J Med. 2020;383:1305–1316) was a multicenter investigator-initiated RCT comparing early rhythm control with usual care in almost 3,000 patients with early AF (diagnosed ≤1 year before enrollment) in terms of composite of cardiovascular death, stroke, or hospitalization with worsening of heart failure or acute coronary syndrome (first primary outcome). The trial was stopped for efficacy at the third interim analysis after a median of 5.1 years of follow-up per patient with a first-primary-outcome event occurring in 3.9 and 5.0 per 100 person in patients assigned to rhythm control vs standard strategy respectively (hazard ratio, 0.79; 96% confidence interval, 0.66 to 0.94; P=0.005).  In particular, a significant reduction of stroke was observed in rhythm control strategy (0.6% vs. 0.9%; HR: 0.65; 95%CI: 0.44 - 0.97). Similar results with pharmacological rhythm control have been observed. The ATHENA trial (Hohnloser SH et al. N Engl J Med 2009;360:668-78), a multicenter, double-blind, placebo-control RCT comparing dronedarone vs placebo in 4628 AF patients in terms of first hospitalization due to cardiovascular events or death (primary endpoint), showed after a mean follow-up of almost 2 years a significant reduction of primary endpoint in the rhythm control group (31.9% vs. 39.4%; HR: 0.76; 95%CI: 0.69-0.84; p<0.001). Of note, a post-hoc analysis of this RCT showed that Dronedarone significantly reduced the stroke risk (HR: 0.66, 95%CI: 0.46 to 0.96, P=0.027). The reason why these two trials unlike the previous ones gave positive results may include different aspects: study design/size, timing and type of rhythm control intervention with different percentage of rhythm control success, etc. It is possible that the early restoration of sinus rhythm observed in these two studies could have possibly prevented irreversible atrial changes and reduced the risk of stroke.

In the paragraph “Rhythm control” we deliberately decided to focus on AF patients with history of stroke only without describing the effects of rhythm control in the overall AF population. That is the reason why we decided to report only the RAFAS Study (the only RCT assessing the effect of early rhythm control among AF patients with recent stroke).

Based on the above evidence (limited to overall AF patients with or without history of stroke), the RAFAS Study results, and the recent State-of-the-Art, we believe that rhythm control strategy deserves to be considered in clinical practice as potential strategy to further reduce stroke in patients experiencing breakthrough stroke.

However, we completely agree with the Reviewer about the lack of balance in describing this stroke reduction strategy. We therefore decided to add/change the following sections of the manuscript:

  • Line 433 “The evidence about the benefit of this strategy on reducing stroke risk in the overall AF population is still controversial (Camm AJ et al. J Am Coll Cardiol 2022;79:1932–1948). However, very few studies have so far investigated the effects of early rhythm control in AF patients with recent ischemic stroke”.
  • Line 444: However, it is important to underline several aspects: first, patients older than 80 years, or with life expectancy of <12 months, or with large left atrium (diameter >55 mm), or with contraindication to OAC or with recent/planned acute coronary syndrome/cardiac inter-vention were excluded from the study; second, the study was not double-blinded and we cannot exclude that the early rhythm group received more aggressive drug treatment to prevent stroke recurrence; third, data regarding to the use of OAC at the time of the is-chemic event were not reported(57).”.

We think that the new version of the manuscript better describes pros and cons of each strategy. However, we remain entirely open for suggestions regarding how the text should be further modified.

Reviewer 2 Report

This is a very elegant and interesting study. The population of patients who experience stroke despite the use of oral anticoagulation is difficult and troublesome for practitioners. The paper inn details describes potential reasons, diagnostic strategy and treatment. Congratulations 

Author Response

We would like to thank the reviewer for his/her overall positive reception of our manuscript and for the very supportive comment.